# Non-Enzymatically Colorimetric Bilirubin Sensing Based on the Catalytic Structure Disruption of Gold Nanocages

**DOI:** 10.3390/s23062969

**Published:** 2023-03-09

**Authors:** Wenxiang Xiao, Yinan Xiong, Yaoxin Li, Zhencheng Chen, Hua Li

**Affiliations:** 1School of Life and Environmental Sciences, Guilin University of Electronic Technology, Guilin 541004, China; 2Guangxi Colleges and Universities Key Laboratory of Biomedical Sensing and Intelligent Instrument, Guilin University of Electronic Technology, Guilin 541004, China

**Keywords:** gold nanocages, bilirubin detection, nanoenzyme, structural collapse, colorimetric sensing

## Abstract

As an essential indicator of liver function, bilirubin is of great significance for clinical diagnosis. A non-enzymatic sensor has been established for sensitive bilirubin detection based on the bilirubin oxidation catalyzed by unlabeled gold nanocages (GNCs). GNCs with dual-localized surface plasmon resonance (LSPR) peaks were prepared by a one-pot method. One peak around 500 nm was ascribed to gold nanoparticles (AuNPs), and the other located in the near-infrared region was the typical peak of GNCs. The catalytic oxidation of bilirubin by GNCs was accompanied by the disruption of cage structure, releasing free AuNPs from the nanocage. This transformation changed the dual peak intensities in opposite trend, and made it possible to realize the colorimetric sensing of bilirubin in a ratiometric mode. The absorbance ratios showed good linearity to bilirubin concentrations in the range of 0.20~3.60 μmol/L with a detection limit of 39.35 nM (3σ, *n* = 3). The sensor exhibited excellent selectivity for bilirubin over other coexisting substances. Bilirubin in real human serum samples was detected with recoveries ranging from 94.5 to 102.6%. The method for bilirubin assay is simple, sensitive and without complex biolabeling.

## 1. Introduction

Bilirubin is a pigment in human bile. It is also an active substance produced by the catabolism of hemoglobin. Bilirubin is present in the serum in various forms, such as free state, glucuronide conjugation or reversibly binding to serum proteins [1]. The normal serum total bilirubin concentration in healthy individuals ranges from 3.4 to 17.1 μM, and is 10 times higher in newborns than in adults. Impaired bilirubin metabolism is very common in newborns and may lead to jaundice and yellowing of the skin and other tissues [2,3]. Bilirubin concentrations in the blood above 170 μM might trigger liver and biliary tract dysfunction, even cause irreversible damage to the brain and nervous system [4,5]. Low bilirubin levels indicate iron deficiency (anemia) and may be directly linked to diabetes, coronary heart disease and cardiovascular disease [6,7,8]. Therefore, bilirubin detection is of clinical importance for diagnosis and treatment of related diseases [9,10].

The clinical methods commonly used to detect bilirubin levels are mainly colorimetric methods based on diazonium reagents [11,12]. The disadvantages of this method are that the reaction is time-consuming and highly pH-dependent [13,14]. Direct spectrophotometry is susceptible to interference from other pigments, limiting its effectiveness. Chromatographic and capillary electrophoresis methods have high selectivity towards bilirubin. Their disadvantages are expensive instruments, complicated operations and time-consuming effort [15]. The oxidase method is highly sensitive and has substrate specificity [16,17,18]. Enzymes such as bilirubin oxidase, peroxidase or nanomaterial-based mimetic enzymes are commonly available to catalyze the oxidation of bilirubin. Natural enzymes own high catalytic activity, but they are sensitive to media physicochemical conditions. Nanozymes exhibit inherently high durability to extreme reaction conditions [19]. In 2007, Yan’s group first found that magnetic Fe_3_O_4_ nanoparticles exhibit peroxidase-like activity [20]. Since then, nanomaterials with enzymatic activity have been widely reported and used in biosensing, bioassay and biomedical fields [21,22,23,24]. Pranab reported a colorimetric method for the detection of free bilirubin based on the peroxidase-like catalytic activity of gold nanoclusters [25]. Li et al. prepared a non-enzymatic electrochemical sensor based on the high catalytic activity of cerium nanocubes, which could detect free bilirubin in a short response time [26]. Therefore, nanozymes would be a strong candidate for natural enzymes to bilirubin oxidation [27,28]. Gold nanocages (GNCs) are hollow nanocrystals with porous walls that possess excellent optical, chemical and catalytic properties [29,30,31,32]. In general, there are few examples of gold nanocages discussed for sensing applications, compared with the field such as photothermal therapy. The nanocages could be used as the substrate for enzyme immobilization to improve the biocatalytic activity of urate oxidase or glucose oxidase and facilitate enzymatic biosensor fabrication [33,34]. Graphene oxide-gold nanocage hybrid has been applied as a probe of surface-enhanced Raman spectroscopy (SERS) for the label-free nitro explosives identification at femto-molar levels through donor-acceptor interactions [35]. Except for intermolecular recognition, most sensing applications were based on the catalytic effect of GNCs. GNCs have been proven to possess peroxidase-like activity. Zeng et al. compared in detail the catalytic activities of three types of gold-based nanostructures and found that gold nanocages are catalytically more active than nanoboxes and solid nanoparticles [36]. The increase in the catalytic activity of GNCs can be ascribed to their hollow particle structure that provides higher and accessible catalytic sites and the thin but electrically continuous walls that allow electrons to transport quickly across the entire surface of an Au nanocage. Au nanocages have been reported as artificial nanoenzymes for the electrochemical sensing of H_2_O_2_, methanol and glucose [37,38,39]. Colorimetrically, catalytic sensing based on gold nanocages often requires the substrate solution to behave sensitive to color changes or the involvement of a third-party dye such as 3,3’,5,5’-tetramethylbenzidine (TMB) [36,40].

Herein, we propose a colorimetric method for bilirubin detection based on GNCs in a ratiometric mode (Figure 1). Differing from GNCs prepared by the galvanic replacement reaction [36], the nanocages were prepared by using hexamethylenetetramine as a template, where the walls are not continuous but composed of Au nanoparticles. The synthesized GNCs have a surface plasmon resonance (SPR) peak around 808 nm and can directly accelerate bilirubin oxidation. During the catalysis, the cage structure of GNCs is partially disintegrated into free gold nanoparticles. Bilirubin oxidation events are transduced into the optical signals of surface plasmon resonance of GNCs and gold nanoparticles. The absorbance ratios of GNCs and Au nanoparticles SPR peaks exhibit good linearity to bilirubin concentrations. This work provides a paradigm for understanding the relationship between the catalytic properties and the structure of nanocages. The bilirubin sensing is label-free, highly sensitive and does not require additional dyes to provide signals. The practical validity of the developed method was assessed by the assay of BR in real human serum samples.

## 2. Materials and Methods

### 2.1. Apparatus and Reagents

The reagents used in the experiments were all analytical grades. Chloroauric acid (HAuCl_4_·3H_2_O), silver nitrate (AgNO_3_), L-ascorbic acid (AA), sodium hydroxide (NaOH), poly (N-vinyl-2- pyrrolidone) (PVP, K30) and hexamethylenetetramine (HMT) were purchased from Sigma-Aldrich (St. Louis, MO, USA). Bilirubin (BR) was obtained from Shanghai Macklin Reagent Company. Bilirubin (1.2 mg) was first dissolved with 0.02 M NaOH and then diluted with 50 mM phosphate-buffered solution (PBS) (pH 7.4) to 10 mL to obtain a stock solution (2.00 × 10^−4^ M). The water used for the experiments was ultrapure (18.2 MΩ·cm).

The UV-vis absorption spectra were measured with a Hitachi (Tokyo, Japan) UH-5300 spectrophotometer. The morphology of the GNCs was observed on a JEM 2100F transmission electron microscope (TEM) at an accelerating voltage of 200 kV. The particle size distribution/zeta potential was obtained from a Zetasizer Nano ZS90 (Malvern, UK) particle size analyzer (DLS).

### 2.2. Preparation of Gold Nanocages

For the synthesis of gold nanocages, HAuCl_4_ (100 μL, 20 mM) and HMT (4.0 mL, 0.03 M) were added to a 10.0 mL beaker at room temperature under stirring. The color of the HAuCl_4_ solution changed from pale yellow to colorless. After stirring vigorously for 2 min, 3.0 mL of 0.30 M PVP and 120 μL of 0.01 M AgNO_3_ were added to the solution. The color did not change at this point and remained colorless. Then, 50.0 μL 0.08 M AA was added. The solution was continuously stirred for 20 min. The solution color was observed to change gradually from colorless to lavender and eventually turn into dark purple as the stirring time was extended. Then, the solution was allowed to react without agitation at room temperature for 12 h. The resultant solution was centrifuged (10 min, 10,000 rpm) and washed twice with ethanol and ultrapure water, respectively, to remove the unreacted components and gold nanoparticles that were not involved in forming the cage structure. Finally, the residue was redispersed in 5.0 mL ultrapure water and stored at 4 °C.

### 2.3. Determination of Bilirubin

Five microliters of bilirubin with a certain concentration was added to 1.0 mL GNCs solution and equilibrated for 10 min. Then, the plasma resonance absorbance was recorded at 502 nm and 808 nm before and after bilirubin addition in the wavelength range of 400~950 nm. Bilirubin was quantified by the absorbance ratios of the two peaks versus concentrations.

## 3. Results and Discussion

### 3.1. Preparation and Characterization of Gold Nanocages

This work aims to develop a method to quantify serum bilirubin using gold nanocages as catalysts and sensing materials. The tunable LSPR peak of GNCs enables them to act as robust optical sensors. First, GNCs have hollow cage-like structures with cavity-limiting effects. A large specific surface area and absorption cross section allow them to catalyze the reaction of bilirubin in solution more readily. Secondly, the nanocage formed by the dipole-dipole gravitational aggregation among nanoparticles is relatively stable. The cage structure has a specific strength and is not easily disturbed. Finally, GNCs synthesis can be quickly and safely finished at room temperature.

The GNCs were synthesized by a one-pot method, in which AA was used as a reducing agent to reduce the precursor HAuCl_4_ to gold nanoparticles. AgNO_3_ acted as an auxiliary agent [41]. AuNPs grew along a complex template formed by HAuCl_4_ and HMT to aggregate into cage structures. The color of the resultant solution was purple-black, depending on the particle size and structure of the GNCs. TEM images show that the GNCs are square nanostructures with hollow interiors (Figure 1a). They offer dark edges and bright interiors, similar to the results previously reported in the literature [42]. The DLS result shows a uniform particle size distribution with an average size of 81.29 nm. In the visible range, the absorbance spectrum has a weak LSPR band centered at 502 nm for AuNPs. In the NIR range, the broad and strong LSPR band near 808 nm is ascribed to GNCs (Figure 1b). The surface of GNCs is negatively charged with a zeta potential of −19.80 mV. 

### 3.2. Optimization of the Preparation Conditions for Gold Nanocages

For the one-pot preparation of GNCs under different conditions, the nanocages may have different LSPR bands and catalytic effects. For example, adjusting the molar ratio between HAuCl_4_ and HMT may change the size of the framework complex so that the LSPR peak can be tuned to the near-infrared region. The main experimental factors, such as the molar ratio of HAuCl_4_ to other reactants and the reaction time during preparation, were optimized to obtain GNCs with higher yield and catalytic efficiency. Here, the ratio of the peak absorbance for GNCs (A_1_, 808 nm) versus the maximal absorbance of AuNPs (A_0_, 502 nm) was used to characterize the synthesis of GNCs. The higher value of A_1_/A_0_ indicates the higher yield of GNCs during synthesis.

The formation of GNCs involves three processes. First, the color of the solution changed from yellow to transparent after adding HMT to HAuCl_4_, indicating the formation of a complex with a dimensional structure. After that, PVP and AgNO_3_ were added. The role of PVP is to stabilize the high-energy surface of the nanoparticles to prevent them from aggregating and support the cage framework [43]. Silver ions can be used as an underpotential deposition agent to access different particle shapes by controlling the growth of the resulting AuNPs through surface passivation. Sedimentation is critical for the subsequent formation of cage structures. Finally, the GNCs started to aggregate and gradually connect along the structural complex under dipole-dipole interaction.

The molar ratios between the reactants, i.e., HAuCl_4_ to HMT and AgNO_3,_ were investigated. The results are shown in Figure 2. Figure 2a demonstrates that the peaks of AuNPs are blue-shifted, and the peaks of GNCs are red-shifted as the ratio of HMT increases. GNCs with NIR resonance absorption peak can be obtained when the molar ratio increases to 45 because HMT and Au^3+^ generate two-dimensional coordination polymers with square cavities. The yield of GNCs (A_1_/A_0_) increases linearly with increasing HMT concentration, reaching a maximum at 60 and decreasing after that. As depicted in Figure 2b, the A_1_/A_0_ ratio rises with the increase of the AgNO_3_ ratio and arrives at its highest when the molar ratio is 0.6. Due to the potential deposition effect of Ag atoms, adequate silver ions are beneficial to grow the HMT-Au^3+^ complex into a more stable cubic structure. Further increase of AgNO_3_ would hinder the growth of gold nanoparticles, resulting in a blue shift of their LSPR peaks, which is unfavorable for subsequent experiments. Therefore, the molar ratios of HAuCl_4_:HMT and HAuCl_4_:AgNO_3_ were set at 1:60 and 1:0.6 for the synthesis of GNCs.

As illustrated in Figure 2c, A_1_/A_0_ values firstly increase and remain stable with the rise of temperature. The GNCs solution gradually changes from purple-black to green-black (inset of Figure 2c). Maximum absorption wavelength of AuNPs can be observed to blue-shift from 502 nm to 476 nm. It may be due to the faster reaction and the quick nucleation at higher temperatures, leading to a decrease in particle size. Further increasing the temperature would reduce the yield of GNCs due to the destabilization of the interparticle structure at high temperatures. The rise in ambient temperature leads to an increase in the kinetic energy of the particles, which is sufficient to overcome the energy barrier responsible for electrostatic stability [44]. Therefore, the temperature range of 10~30 °C is suitable for GNCs preparation.

As shown in Figure 2d, A_1_/A_0_ ratio goes up whit prolonged reaction time. A proper extension of time will make the reaction complete more thoroughly. Until the reaction time extends to 20 min, A_1_/A_0_ reaches the maximum and then declines. Further extension of the reaction time may be detrimental to the aggregation of nanoparticles, leading to a decrease in the concentration of GNCs.

### 3.3. Catalytic Oxidation of GNCs on Bilirubin, the Oxidative Activity and the Underlying Mechanism

Unlike traditional bio-enzymes, nanoenzymes are less susceptible to interference from external factors and have better stability. Since gold nanocages are stable gold-based nanomaterials with oxidase-like activity, this material has been selected as enzymatic mimetics for quantifying bilirubin. The cage structure endows GNCs a cavity-limiting effect that enhances the catalytic activity. Bilirubin is a special endogenous antioxidant with a bright orange color that can be oxidized to colorless or dark green pyrrole-like substances. The interaction between GNCs and bilirubin was investigated, and the LSPR absorption spectra are shown in Figure 3a. Apparently, the peak intensity of AuNPs (A_0_, 502 nm) rises and the intensity of GNCs (A_1_, 808 nm) decreases compared to that of GNCs without bilirubin. The variation in LSPR peak intensities laid a foundation for the determination of bilirubin.

The peak intensity changes of GNCs induced by bilirubin exposure may be related to the catalytic oxidation of bilirubin. To confirm whether GNCs could catalyze bilirubin, the mixed solution of GNCs with bilirubin (C_bilirubin_ = 10 µM) was incubated at 37 °C for a certain time. Then the solution color and UV-Vis spectra of bilirubin were recorded. As time passed, the color of the solution faded (inset of Figure 3b), and the characteristic absorbance of bilirubin at 440 nm declined (Figure 3b), indicating the degradation of this substrate. Moreover, a new weak peak appeared at 600~700 nm, which is ascribed to the absorption of biliverdin. These results demonstrate that GNCs exhibit bilirubin oxidase-like activity. 

The apparent kinetics parameters such as the Michaelis–Menten constant (K_M_) and maximum initial velocity (V_max_) were measured by using the Lineweaver–Burk model to assess the catalytic activity of GNCs for bilirubin [45]. The K_M_ and V_max_ were calculated to be 2.07 × 10^−2^ M and 4.60 × 10^−9^ M s^−1^, respectively (Figure 3c). The lower K_M_ value implies the high affinity of GNCs towards bilirubin. According to the theory of enzymatic catalysis, K_M_ represents the affinity of a given enzyme toward the substrate determined by its use. The initial velocity (v) of the oxidation of bilirubin was detected from the decreasing concentration of bilirubin versus time, which was calculated from the absorbance using an average molar extinction coefficient of 47,500 M^−1^ cm^−1^ determined from the linear plotting of the absorbance versus time obtained by Michaelis–Menten equation. Due to the folded structure of bilirubin and the adsorption of GNCs, bilirubin aggregates easier, thus the oxidation of bilirubin may be accelerated merely by the trace amount of oxygen dissolved in water. The catalytic activity of the nanoenzyme was determined by measuring the slope of the initial linear portion of the nanocages reaction curve [45]. All these results suggest that GNCs exhibit oxidase-like activity. Without the need for complex biomarkers, aerobic bilirubin oxidation is catalyzed, and oxidation events are transduced into the surface plasmon resonance signals of GNCs and gold nanoparticles. 

As illustrated in Figure 3d, compared with GNCs (Figure 1a), TEM images of the GNCs-bilirubin system show that the smooth edges of GNCs become rough after BR addition. It proves that GNCs have a catalytic effect on bilirubin and bilirubin induces structural collapse of GNCs. Bilirubin prefers to adopt the energy minimum conformation, that is a ridge-tile conformation stabilized by intramolecular hydrogen binding [46]. The structure may be easier to insert into the space between the gold nanoparticles, enabling GNCs to be selective for bilirubin. Moreover, the interaction between the gold nanoparticles and the polar groups such as carboxyl and hydroxyl in the bilirubin molecule weakens the dipole forces among the nanoparticles and damages the aggregation of AuNPs on nanocages. Therefore, we hypothesize that the dipole-dipole gravitational force between nanoparticles on the nanocage is disrupted by GNCs-bilirubin catalysis releasing the AuNPs from the cage structure. Thus, a ratiometric colorimetric method for bilirubin can be established based on the structural collapse of GNCs induced by bilirubin.

### 3.4. Optimization of Experimental Conditions for Bilirubin Catalysis

Based on the catalytic effect of GNCs, a sensor has been developed to detect bilirubin rapidly. The concentration of GNCs, pH value, temperature and the reaction equilibrium time may affect the catalytic impact of GNCs on bilirubin. Therefore, the above experimental factors were optimized to obtain better catalytic effects.

Different concentrations of GNCs on bilirubin catalysis were investigated by diluting GNCs with ultrapure water. As shown in Figure 4a, the A_0_/A_1_ change of GNCs in the absence of bilirubin is insignificant with the enlargement of the dilution factors. Therefore, dilution does not affect the structure of GNCs. After bilirubin addition, the A_0_/A_1_ ratio decreases slightly when using diluted GNCs solution. When the dilution factor exceeds 1.4, the A_0_/A_1_ and the difference between the A_0_/A_1_ values before and after the addition of bilirubin (△(A_0_/A_1_)) rise with the enlargement of the dilution factor, and △(A_0_/A_1_) reaches the maximum value at 1.8. Under a lower dilution factor, the GNCs are more evenly dispersed, and their larger specific surface area allows the GNCs to fully contact with bilirubin, thus enhancing the catalytic effect on bilirubin. With further increase in the dilution ratio, △(A_0_/A_1_) drops. In the case of excessive dilution, the concentration of GNCs decreases and the catalytic effect on bilirubin is diminished. A dilution factor of 1.8 is optimal.

The pH of the solution may affect the catalytic reaction of GNCs to bilirubin, and a suitable pH value is beneficial to improve the sensitivity of the assay. As illustrated in Figure 4b, the GNCs are relatively stable at pH 6.0–7.0 before the addition of bilirubin. Once the pH value exceeds 7.4, the A_0_/A_1_ value increases with elevating media pH, but the wavelength of GNCs rapidly red-shifts. The reason may be attributed to the charged property of GNCs that are changed under alkaline conditions, resulting in structure destabilization. After adding bilirubin, the value of △(A_0_/A_1_) also elevates when the pH value rises from 5.0 to 6.8 and reaches the maximum at pH = 6.8. After that, △(A_0_/A_1_) drops as the pH value increases. The structure of GNCs is probably disrupted under alkaline conditions, resulting in a decrease in catalytic capacity. It can be concluded that GNCs react more efficiently with bilirubin in a weakly acidic media.

The effect of temperature on the catalytic reaction of GNCs on bilirubin was explored in the range of 25~47 °C, and the results are shown in Figure 4c. In the absence of bilirubin, the fluctuation of A_0_/A_1_ with increasing temperature is minor. In the presence of bilirubin, A_0_/A_1_ gradually increases and reaches the highest at 37 °C. Further increase in the temperature gives a decreasing trend for both A_0_/A_1_ and △(A_0_/A_1_). Higher temperatures can speed up the response and may allow the catalytically disintegrated structures to rejoin each other due to the dipole-dipole gravity of the nanoparticles. Therefore, the sensor has the best response sensitivity to bilirubin at 37 °C.

As shown in Figure 4d, the A_0_/A_1_ value of the GNCs-bilirubin system increases with the extended reaction time and reaches a maximum at 4 min. After 4 min, the A_0_/A_1_ value decreases slightly and remains stable after 10 min. Therefore, the catalytic reaction and cage structure disintegration between GNCs and bilirubin require 10 min to arrive at dynamic equilibrium.

### 3.5. Analytical Performance of Gold Nanocages for Bilirubin Detection 

To establish a quantitative relationship between the structural disintegration of GNCs and bilirubin concentration, GNCs interacted with different concentrations of bilirubin under optimal experimental conditions. The absorption spectra are shown in Figure 5a. The local plasmon resonance peak of AuNPs is around 500 nm, and the maximum wavelength of bilirubin is near 440 nm. The absorption spectra of the two substances overlap each other to some extent. The maximum absorption wavelength of AuNPs blue-shifts to between 440 nm and 500 nm before the catalytic reaction. However, the LSPR peak of AuNPs returns to about 500 nm, after GNCs interact with bilirubin and the bilirubin absorbance disappears. This fact proves that catalytic oxidation really occurs between GNCs and bilirubin, while the peak wavelengths of GNCs do not change with bilirubin addition. As mentioned above, the dipole-dipole gravity among gold nanoparticles on the nanocages can be destroyed by the bilirubin catalysis, and free AuNPs are released from the nanocages. Thus, an increase in the absorbance of AuNPs can be observed due to more AuNPs liberating from the cage with increasing bilirubin concentration, while the absorbance of GNCs decreases with the disintegration of the cage structure. When the bilirubin concentration further increases, the LSPR peak of AuNPs blue-shifts toward 440 nm, which indicates that the catalytic effect of GNCs reaches saturation. The absorbance of GNCs no longer descends linearly with enlarging bilirubin concentration.

Bilirubin quantification was performed by subtracting the value of the black control (A_0_/A_1_) in the absence of bilirubin. A linear relationship between the A_0_/A_1_ difference of the GNCs-bilirubin system is achieved in the bilirubin concentrations range of 0.20~3.60 μmol∙L^−1^ (Figure 5b). The fitted equation for A_0_/A_1_ versus C_bilirubin_ can be expressed as A_0_/A_1_ = 0.0526C (μmol∙L^−1^)–0.0033, with a correlation coefficient of R^2^ = 0.99. The limit of detection is calculated to be 39.35 nM (*S/N* = 3, *n* = 3). The relative standard deviation (RSD) for five replicate measurements of 0.20 μmol∙L^−1^ of bilirubin was 0.095%, indicating good reproducibility of the proposed method.

### 3.6. Specificity of Gold Nanocages for Bilirubin Detection

To evaluate the selectivity of GNCs towards bilirubin, different biomolecules (glucose, fructose, dopamine, phenylalanine, galactose, glutathione, cysteine, urea, histidine and cholesterol) and several common metal cations (Zn^2+^, K^+^, Na^+^ and Ca^2+^) (50.00 μmol/L each) were investigated. Each biomolecule was coexisted with 1.00 μmol/L of bilirubin and analyzed according to the above procedures. Figure 6 shows that these coexisting components do not interfere with the detection of bilirubin, indicating that the GNCs has good selectivity towards bilirubin.

The comparisons of the analytical parameters, such as linear range and limits of detection for bilirubin based on various analytical processes, are shown in Table 1. This proposed method has a relatively rapid response to bilirubin. Both its detection limit and linear range are comparable with the other reported methods.

### 3.7. Determination of Bilirubin in Human Serum Samples

Bilirubin concentrations in human serum samples were measured to evaluate the applicability of the method. Serum samples were provided by a local hospital in Guilin, China, and serum bilirubin concentrations were determined using the standard addition method. Serum (10.0 μL) was added to 1.0 mL of GNCs test solution, equilibrated for 10 min, and the absorbance was measured. In addition, a known concentration of standard bilirubin solution was spiked, and the absorbance was recorded. The bilirubin concentrations and recoveries were calculated and shown in Table 2. The spiked recoveries are in the range of 94.5~102.6%, indicating the excellent applicability of the method.

## 4. Conclusions

A novel colorimetric sensor based on the catalytic disintegration of GNCs structure has been developed for bilirubin detection. GNCs with an internally hollow square cage-like structure have been used as the catalyst for bilirubin oxidation. The nanocage material presents double localized plasmon resonance peaks at visible and NIR region, corresponding to gold nanoparticles and nanocages, respectively. The catalytic oxidation of bilirubin triggers the disintegration of the cage structure by disrupting the dipole-dipole gravitational forces between nanoparticles and releasing free AuNPs into the solution. Thus, the peak absorbance for AuNPs increases with bilirubin exposure accompanied by the descending of the GNCs peak. Ratiometrically, colorimetric sensing of bilirubin has been achieved based on the linear relationship between the ratio of two peak absorbances versus bilirubin concentrations. The bilirubin sensing is label-free, highly sensitive, and satisfactory results were obtained in serum tests. This work provides insights for sensor fabrication based on the relationship between the catalytic properties and the structure of nanocages, signifying the potential of GNCs as biosensors in the point-of-care testing of BR as a liver function marker. In the future work, the sensing method might be combined with 3D printing technology to build a portable instrument system [54,55].

## Data Availability

Not applicable.

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
