# Peer review of "Non-Enzymatically Colorimetric Bilirubin Sensing Based on the Catalytic Structure Disruption of Gold Nanocages"

_sensors, 2023, doi:10.3390/s23062969_

Round 1

Reviewer 1 Report

the paper is well written and organized, the experimental design is complete and all the experiment necessary to demonstratate the significance of the methods are reported.

for this reason I recommend publication of this paper after minor revision.

Only two suggestion for the authors:

1. in the introduction is necessary to indicate the physiological and pathological level of bilirubin in blood. this aspect can be important to compare the  it with the concentration range test in the present work

2. please add the error bars in the calibration curves in fig 5

Reviewer 2 Report

Since bilirubin detection is of clinical importance for diagnosing and treatment of related diseases, the authors proposed a colorimetric method for bilirubin detection based on gold nanocages in a ratio metric mode. The authors discuss preparation, optimization, and characterization of gold nanocages. Finally the authors provide discussion for performance of gold nanocages for bilirubin detection and determination of bilirubin in human serum samples. The article is organized well, and it is structured in a good shape. However, there are major issues that should be addressed prior to possible publication. The following points are strongly suggested to be addressed:

1.      Analytical performance of a biosensor is the most important and crucial part of a sensing platform. In this project (Fig. 5a), why the Y axis (Abs) started from around 0.45 instead of 0.00?

2.      In Fig. 5b, the error bars seem unrealistically small. How did the authors calculate the error bars? Please provide the formula and explanation for it.

3.      In section 3.6, the authors used 1 μmol/L concentration for selectivity studies. Considering Fig. 5b (calibration studies), the sensor response for such concentration should be around 0.05 while in the fig. 6, the presented result looks more than 0.055 towards 0.06. Considering the reported error bars, how do the authors justify this?  

4.      Abstracts should contain the most key parts and important results and it should be short/informative/attractive. Please revise the abstract.

5.      It is suggested to improve the structure of the introduction by summarizing some of the information and presenting 3 cohesive paragraphs with the first two emphasizing the importance of the work/literature and the last one describing a gist of the work.

6.      Units in Table 1 are not unified to offer an appropriate comparison of the presented works (some are described in μM and some in nM). Either the units should be unified, or the table should be sorted in a meaningful order (such as smallest to largest).

7.      It is suggested that the authors expand the conclusion section by proving more insights for future perspectives such as potential implementing of artificial intelligence for optimizing fabrication/detection. Some helpful references can be: doi.org/10.3390/bios12070491 and doi.org/10.3390/mi13071099

8.      It is suggested to proofread the article for small typos and/or inconsistent styling. For example, in several places punctuation marks are missing (lines 26, 38, 201, and so on). Another example is double labeling in Figure 1.

Round 2

Reviewer 2 Report

The authors addressed all of the comments.